Comparison of supervised versus home-based exercise programs following platelet-rich plasma injections in knee osteoarthritis: clinical outcomes from quasi-experimental study

Safdar Ghalia 1 2
Kiyani Mubin Mustafa mubin3us@yahoo.com 2 3
Saleem Salman Ahmed 2
Qureshi Wajeeha Irfan 4
Bashir Shahid 5
Abualait Turki 6
Khan Hamid 7
Sajjad Abdul Ghafoor 8
1 Department of Rehabilitation Sciences, Shifa Tameer-e-Millat University , Islamabad , Pakistan
2 Shifa International Hospitals, Ltd , Islamabad , Pakistan
3 Shifa College of Medical Technology, Shifa Tameer-e-Millat University , Islamabad , Pakistan
4 Riphah International University , Islamabad , Pakistan
5 Neuroscience Center, King Fahad Specialist Hospital , Dammam , Saudi Arabia
6 College of Applied Medical Sciences Eastern Campus, Imam Abdulrahman Bin Faisal University , Dammam , Saudi Arabia
7 International Islamic University , Islamabad , Pakistan
8 Islamabad Institute of Allied Health Sciences, Dr Akbar Niazi Teaching Hospital , Islamabad , Pakistan
Mourão Carlos Fernando
Electronic publication date: 2025 Oct 14
Publication date: 2025
Volume: 13
Electronic Location ID: e19971
Received 2025 Mar 14; Accepted 2025 Jul 31
Copyright: ©2025 Safdar et al.
Copyright year: 2025
Copyright holder: Safdar et al.
License: This is an open access article distributed under the terms of the Creative Commons Attribution License, which permits unrestricted use, distribution, reproduction and adaptation in any medium and for any purpose provided that it is properly attributed. For attribution, the original author(s), title, publication source (PeerJ) and either DOI or URL of the article must be cited.
License URL: https://creativecommons.org/licenses/by/4.0/

Keywords: Home-based exercise, Knee osteoarthritis, Pain management, Platelet-rich plasma (PRP), Range of motion (ROM), Supervised exercise

Funding: King Salman Center For Disability Research KSRG-2024-307 This work was funded by the King Salman Center For Disability Research through Research Group no KSRG-2024-307. The funders had no role in study design, data collection and analysis, decision to publish, or preparation of the manuscript.

==============================
Background

Knee osteoarthritis (OA) is characterized by severe pain and decreased range of motion (ROM), mostly managed with conservative treatments. Therapies like platelet-rich plasma (PRP) intra-articular injections have gained fame for pain relief when surgery is a last resort.

Objectives

To compare the effectiveness of a supervised exercise program versus a home-based exercise program following PRP injections in terms of pain relief, ROM improvement, and disability reduction in knee osteoarthritis patients.

Methods

A quasi-experimental design was employed, enrolling 32 patients diagnosed with knee OA cases (age range: 40–60 years; 59% female), divided into two groups: a supervised exercise group (n = 16) and an unsupervised/home-based exercise group (n = 16). Both groups received two PRP injections spaced 15 days apart. The supervised group followed a structured exercise protocol, including cold packs post-injection for pain management. The program comprises three sessions per week for one month. The unsupervised group received similar exercises to be performed independently at home. Data analysis was conducted using parametric and non-parametric statistical methods based on data distribution.

Results

Both groups exhibited significant improvements from baseline to post-intervention in pain, ROM, and functional status (p < 0.05). The supervised exercise group demonstrated superior outcomes in pain reduction, knee flexion ROM, and Knee Injury and Osteoarthritis Outcome Score (KOOS). However, no significant difference was observed between groups for knee extension ROM (p = 0.378).

Conclusion

Both supervised and unsupervised exercise programs improved outcomes in knee OA patients following PRP injections; the supervised program showed greater pain relief, flexion ROM, and functional status. However, no significant difference was observed in knee extension ROM between the two groups.

Introduction

Knee osteoarthritis (OA) is one of the most prevalent musculoskeletal disorders of joints, primarily affecting the old age population. The main features of OA is the slowly progressive degeneration of articular cartilage, which results in pain chronicity, muscle stiffness, reduced range of motion (ROM) and functional disability of the knee joint. This degenerative osteoarthritic illness has a remarkable impact on quality of life of patient, especially main focus to patient mobility as well as physical function (Cross et al., 2014). Women are consider to be more prone to be exposed to OA compared to men due to hormonal, anatomical and biomechanical differences (Srikanth et al., 2005).

Although there are multiple treatment option available to treat OA like multiple pharmacological options such as analgesics, use of nonsteroidal anti-inflammatory drugs (NSAIDs). Some non-pharmacological options are also available to relief the symptoms like lifestyle modification, weight management, physical therapy and certain exercise programs. Traditionally, the management of knee OA has focused on conservative approaches, including pharmacological interventions such as pain reliever analgesics and NSAIDs. Non-pharmacological treatments, including physical therapy, weight management, and lifestyle modification, are also essential in alleviating symptoms. However, as the disease progresses, these traditional interventions seem to be less effective, offering the need of consideration of intra-articular injection therapies like injectable of platelet-rich plasma (PRP), hyaluronic acid (HA), and the novel peripheral blood-derived mononuclear cells (PBMNCs) etc. Out of these, PRP injections have gained popularity and prominence as a new and advance therapeutic option being its regenerative properties (Chiaramonte et al., 2025; Filardo et al., 2015; Filardo et al., 2012).

PRP is extracted from autologous blood and is consisting of a raised concentration of platelets along with richness of growth factors that enhances tissue repair mechanism and regeneration process. The use of PRP is an attractive option in knee OA as it has been reported to reduce pain, improving the joint function and mobilization (Bennell & Hinman, 2005; Bennell et al., 2008). PRP also delay the requirement for surgical interventions. PRP works by controlling the inflammatory response, promoting tissue repair, and enhancing cartilage regeneration. Its less invasive property and highly favourable safety profile make it more catchy option for managing OA (Kohn, Sassoon & Fernando, 2016).

Exercise therapy is also a well-established management of knee OA. The exercise therapy not only involve in improving the strength of muscle, joint mobility but also overall functional capacity of joints like increase in ROM and pain reduction. The combination of PRP with exercise therapy may represents a new emerging area of interest in the management of OA which provide various benefits, especially focus on biological and mechanical aspects of OA (Laudy et al., 2015). In the same context, the mode of exercise delivery is very important whether it is supervised or home based, it’s always a point of discussion. The supervised exercise protocols, most often administered by expert physical therapists, may provide structured guide lines, ensuring the adherence of correct exercise form, duration and intensity. These programs have been shown to improve the therapeutic impact of exercise, primarily focus on patients with more chronic functional disabilities (Fransen et al., 2008; Fransen et al., 2015).

On the other end, home based exercise protocols offer flexible and potentially cost-effective alternatives, which allow independent performance of patients to exercise. Despite so much advantages of home-based programs, it can be still a significant challenge, often leading to suboptimal outcomes. Research comparing the potential efficacy of supervised exercise program with comparison to home based exercise programs in managing knee OA, particularly focus on PRP injections, is less evident but it is important for providing clinical practice guidance (McAlindon et al., 2014; Zhang et al., 2008).

The main aim of the current study is to draw a comparison of the potential of a supervised exercise program with a home-based exercise program following the PRP injections in knee OA patients. Specifically, it assesses the outcomes related to reduction in pain, ROM improvement, and functional disability between these two groups. By providing insight into the most effective post-PRP rehabilitation strategy, this study will also help to optimize rehabilitation protocols for knee OA patients receiving PRP injection.

Materials and Methods

Study design

The current study was a quasi-experimental study designed to analyse the comparative impact of supervised exercise program and home-based exercise program following PRP injections in knee osteoarthritis patients. The study primarily focused on to investigate the differences in pain severity, ROM, and functional disability of joints between the two treatment groups.

Study setting

The participants of the study were recruited at Pain Clinic (Outpatient Department, OPD) of Shifa International Hospital, Islamabad, Pakistan and referred to the physical therapy for the study.

Duration of study

The study was conducted over a time frame of six-months, starting from August 2018 to January 2019.

Sample size

Open Epi software was used for initial sample size calculation which was found as 10 participants per group. Due to the good flow up of patients in the OPD and after taking consent from the supervisor, a total of 32 patients visiting to OPD on regular basis showed willingness of participation, were recruited by purposive sampling method based on patient preference and clinical feasibility and then these were divided into two groups, 16 in the supervised exercise program group and 16 in the home-based exercise program group. Written informed consents were taken from each participant prior to enrolled in exercise program.

Sample selection

The inclusion criteria for the current study comprised the enrolment of the participants based on radiographic confirmation of knee OA (Grades 1–3) using the Kellgren-Lawrence classification system (Kohn, Sassoon & Fernando, 2016) with X-rays independently reviewed by two study physicians. The participants were of 40 to 60 years aged and facing limited range of motion (ROM). Both sexes were eligible to participate. Participants were excluded if they presented with any of the following conditions, verified through comprehensive screening: (1) neurological impairments affecting motor function, assessed via clinical examination for sensory/motor deficits and medical record review; (2) systemic inflammatory diseases including rheumatoid arthritis (confirmed by negative rheumatoid factor and anti-CCP antibody testing); (3) unstable cardiovascular disease, screened through resting ECG and physician evaluation; (4) cognitive impairment (Mini-Mental State Examination score < 24); (5) history of knee trauma, joint infection, or malignancies documented in medical records; (6) radiographic evidence of grade 4 knee OA (Kellgren–Lawrence classification); (7) age outside the 40–60 year range; or (8) inability to comply with the rehabilitation protocol. These criteria ensured the exclusion of confounding conditions that might affect treatment response or exercise participation. All participants reported no participation in structured physiotherapy programs during the 6 months preceding study enrolment.

Measurement tools

Data were collected using three primary instruments. First, A close ended and self-structured questionnaire was design to collect the demographic details and data of initial history of patient. Second a 12-inch stainless steel goniometer (Baseline®, Fabrication Enterprises Inc., Elmford, NY, USA) was applied to take the measurement of knee active ROM which included flexion and extension both at baseline level and after giving intervention. Three trials were recorded per motion, with the mean value used for analysis to minimize variability. Third, the levels of severity of pain in patients were assessed by using the NPRS, which is a segmented numeric version of the visual analogue scale (VAS) (Jensen, Karoly & Braver, 1986). Patients were guided to rate their pain severity on a scale ranging from number 0 indicating no pain to number 10 representing the worst imaginable pain. While the functional outcomes were evaluated using the Knee Injury and Osteoarthritis Outcome Score (KOOS), was implemented for the evaluation of pain, symptoms found, activities of daily living (ADL), sport/recreation as well as knee related quality of life (QOL) (Roos & Lohmander, 2003). This scale shows 42 items within five subscales and the range is scored from 0 to 100, indicating the higher scores as better outcomes.

Primary outcomes included pain intensity, functional status, and knee ROM. The NPRS was used to evaluate pain at baseline level and after the completion of final session (1 month). Functional improvements were assessed using the KOOS. The KOOS was determined at baseline level and after giving 12 sessions for the supervised exercise program group and at the end of the intervention period for the home-based exercise program group. The subscales for measuring pain severity, symptoms found, ADL, sport/recreation activity, and the QOL were analysed separately. The ROM, knee flexion and knee extension were measured by using a goniometer. Normal ROM at the knee ranges starts from 0° of extension at completely straight position of knee joint to 135° of flexion of fully bent knee joint as reported in previous studies (Gogia, Braatz & Rose, 1985).

Ethical considerations

Informed consent was taken from all the subjects before including into the study. The researcher explained the detail of whole procedure to the patients along with benefits and risks like swelling in the knee joint after injection or exercises. Approval from IRB of Shifa International Hospital was taken before commencement of the study under the letter no 1080-356-2018. All the procedure followed the guidelines of Declaration of Helsinki (Puri et al., 2009).

Data collection procedure

The data for the current study was collected at two time points: baseline (before the intervention) as well as at the end of the intervention given (after 12 sessions or 1 month). All the participants recruited in study were divided into two groups using purposive sampling technique based on the study design.

The home-based exercise program group (Group A) followed a home-based exercise protocol while supervised exercise program group (Group B) received PRP injections along with supervised physical therapy sessions. A total of 12 sessions were given to both groups, with the process of data collection at the start of first and ends of last sessions.

Intervention protocols

The CONSORT flow diagram for analysing the progress through the phases of a parallel trial of two groups is given as the Fig. 1 whereas the protocols designed for both groups comprising administration of PRP Injections, application of cold packs, strengthening exercises, stretching exercises, knee mobilization techniques and home exercise plan for the supervised exercise program group and the home-based exercise program group are given in detail in Table 1.

Figure 1 The CONSORT flow diagram.

Table 1 Protocols for both the supervised exercise program group and the home-based exercise program group.

Protocol	Home-based exercise program group (Group A)	Supervised exercise program group (Group B)	
PRP injections	2 PRP injections administered 15 days apart	2 PRP injections administered 15 days apart	
Post-injection cold pack	Cold pack applied for 10–15 min after injection in case of pain	Cold pack applied for 10–15 min after injection in case of pain	
Strengthening exercises	Quadriceps isometrics and vastus medialis obliquus strengthening: 1 set of 10 repetitions, 3 times a day with a 5-second hold for each contraction	Quadriceps isometrics and vastus medialis obliquus strengthening: 1 set of 10 repetitions with a 5-second hold for each contraction	
Stretching exercises	Hamstring and Achilles tendon stretches: 1 set of 10 repetitions, 3 times a day with a 5–10 s hold	Hamstring and Achilles tendon stretches: 1 set of 10 repetitions with a 5–10 s hold	
Knee mobilizations	Not applicable	Anterior tibial glide, posterior tibial glide, patellofemoral medial/lateral/superior/ inferior glides were performed	
Home exercises	Guided for home-based exercise plan: quadriceps and VMO strengthening, hamstring, and Achilles tendon stretching	Home-based exercises guided in addition to supervised sessions (quadriceps/VMO strengthening, hamstring/TA stretching)	
Sessions/Duration	Followed a 1-month home-based exercise plan	3 sessions per week for 1 month (total 12 supervised sessions)	
Notes.

Abbreviations: PRP, platelet-rich plasma; VMO, vastus medialis obliquus; TA, Achilles tendon.

Both groups received identical exercise prescriptions, differing only in supervision level.

The timing and schedule of all interventions, including PRP injections, rehabilitation sessions, and outcome assessments, are detailed in Table 2. Both groups followed identical timelines for injections and evaluations, differing only in exercise supervision. Supervised sessions began 48–72 h post-injection to allow acute post-PRP inflammation to subside, while home-based participants received identical exercises with weekly adherence checks.

Table 2 Treatment timeline and rehabilitation schedule.

Intervention phase	Supervised exercise group (Group B)	Home-based exercise group (Group A)	
PRP injections	Two injections administered:
• Day 0 (baseline)
• Day 15 (±2 days)	Same as Group B	
Rehabilitation sessions	• Initiated 48–72 h post-injection
• 3 sessions/week for 4 weeks (total 12 sessions)
• Session duration: 45–60 min	• Initiated 48–72 h post-injection
• Daily home exercises (same regimen as Group B, but self-guided)
• Weekly phone check-ins	
Cold pack application	Applied post-injection and post-exercise as needed for pain	Applied post-injection only, if needed	
Outcome assessments	• Baseline (pre-injection)
• Post-intervention (Day 30 ± 2)	Same as Group B	
Notes.

Abbreviations: PRP, platelet-rich plasma. Key: All timepoints calculated from first PRP injection (Day 0). Outcome assessments were performed by blinded evaluators.

Data analysis

All the participants were evaluated at baseline and on last session. Continuous data was presented as mean ± standard deviation (SD). Data was entered and analysed through IBM SPSS 22 (IBM Corp., Armonk, NY, USA) and presented in the form of tables and graph. The Shapiro–Wilk test was used to assess normality of the data. Depending on the normality of the data, Independent and Paired t-tests were used as parametric tests whereas Mann–Whitney and Wilcoxon test were done as non-parametric tests or between- and within-group comparisons analysis.Statistical significance was set at p < 0.05 (two-tailed) for all analyses.

Results

The results of this study are based on Objective of this study aiming the comparative analysis between two groups: a supervised exercise program group and a home-based exercise program group, both groups were following the admiration of PRP injections for knee osteoarthritis. The key parameters, including in this study are pain intensity, knee injury, ROM and the Osteoarthritis Outcome Score (KOOS) subscales were analysed pre and post treatment.

Demographics and baseline characteristics

A total of 32 participants having age range of 40 to 60 years were included in the study, with 16 participants in each group. The mean age of the participants was quite similar among the groups, with an average of 53.88 ± 4.68 years in the home-based exercise program group verses 54.75 ± 4.53 years in the supervised exercise program group). Gender distribution among the groups and onset and the duration of pain did not significantly variate among the groups. The two groups had similar baseline characteristics as demonstrated in Table 3 and their differences in age (p = 0.621), gender distribution (p = 0.728), pain levels (NPRS p = 0.782), functional scores (KOOS pain p = 0.421), and range of motion (Flexion ROM p = 0.452) were not statistically significant (all p > 0.05), confirming comparable starting conditions for subsequent intervention comparisons. This will confirm homogeneity of our samples during the pre-intervention.

Table 3 Comparison of baseline characteristics between home-based and supervised exercise groups.

Variable	Home-Based (n = 16) (Mean ± SD)	Supervised (n = 16) (Mean ± SD)	p-value	
Age (years)	53.88 ± 4.68	54.75 ± 4.53	0.621	
Gender (Female: Male)	10:06	9:07	0.728	
Height (cm)	163.7 ± 7.8	165.2 ± 8.1	0.582	
Weight (kg)	76.9 ± 10.2	78.4 ± 9.6	0.673	
BMI (kg/m2)	28.9 ± 3.4	28.7 ± 3.1	0.854	
NPRS (0–10)	6.81 ± 1.28	6.94 ± 1.34	0.782	
KOOS Pain (0–100)	47.50 ± 12.95	51.50 ± 13.58	0.421	
Knee Flexion ROM (∘)	110.50 ± 12.68	113.63 ± 11.41	0.452	
Notes.

Data presented as mean ± standard deviation for continuous variables and counts for categorical data. NPRS, Numeric Pain Rating Scale (higher scores indicate worse pain); KOOS, Knee Injury and Osteoarthritis Outcome Score (higher scores indicate better function); ROM, range of motion. All p-values > 0.05 confirm no significant baseline differences between groups, supporting sample homogeneity at study initiation. Statistical comparisons used independent t-tests for continuous variables and chi-square test for gender distribution.

Aggravating and relieving factors

Aggravating factors for pain comprises the activities like walking (50%), stair climbing (15.6%), low sitting (6.3%), and excessive joint movement (28.1%) while the relieving factors consist of pain relief through rest reported by (46.9%) participants, painkillers (43.8%), and a combination of both rest and painkillers (9.4%) are well expressed in Table 4.

Table 4 Aggravating as well as the relieving factors percentages as per reported by participants.

Aggravating factors: Activities reported to worsen knee pain during daily life. While the relieving factors: Interventions reported to alleviate pain.

S.No	Factors	Frequency	Percentage %	
Aggravating factors				
1	Walk	16	50	
2	Stair climbing	5	15.6	
3	Low sitting	2	6.3	
4	Excessive movement of joint	9	28.1	
Total		32	100	
Relieving factors				
1	Rest	15	46.9	
2	Painkillers	14	43.8	
3	Rest + Painkillers	3	9.4	
Total		32	100	

Time duration of pain onset and behaviour of pain and status of symptoms

The duration of onset of pain variate among participants. Similarly, presence of constant pain was more prevalent resulting into persistent discomfort. Only 6.3% of patients noted relief and any improvement in symptoms. The detail of these features is given in Table 5.

Table 5 Aggravating and relieving factors reported by participants.

Data show most participants had chronic symptoms (>1 year, 43.8%), constant pain (56.3%), and stable disease (68.8%) at baseline.

S.No	Factor	Frequency	Percentage %	
Time duration of onset of pain				
1	Less than 6 months	8	25	
2	6 months to 1 year	10	31.3	
3	More than 1 year	14	43.8	
Total	32	100		
Behavior of Pain				
1	Constant	18	56.3	
2	Intermittent	14	43.8	
Total	32	100		
Status of Symptoms				
1	Improving	2	6.3	
2	Unchanging	22	68.8	
3	Worsening	8	25	
Total	32	100		

Comparative outcomes

There was a remarkably significant variation exist among the home-based exercise program group and the supervised exercise program group with reference to pain, ROM and KOOS scores. The supervised group showed greater improvements across all parameters as indicated in Table 6. It enlightened the post interventional results across different measuring parameters associated to knee OA in both groups, focused on pain intensity, functional outcomes like KOOS, and knee ROM. The supervised group experienced a significantly greater reduction in pain intensity as indicated by NPRS score of 2.44 ± 0.964 with comparison to the home-based group having NPRS score equilant to 4.31 ± 0.873. The p-value < 0.01 represents that the supervised program was highly effective in reducing pain after PRP injection.

Table 6 Comparison of outcomes between the home-based exercise program group and supervised exercise program group.

Between-group outcomes post-intervention.

Variable	Home-based exercise program group (Mean ± SD)	Supervised exercise program group (Mean ± SD)	p-value	
NPRS	4.31 ± 0.873	2.44 ± 0.964	<0.01	
KOOS pain	57.06 ± 8.330	79.44 ± 10.551	<0.01	
KOOS symptoms	62.50 ± 11.967	86.38 ± 11.171	<0.01	
KOOS activities of daily living	52.75 ± 10.98	81.94 ± 8.970	<0.01	
KOOS sports and recreation	23.75 ± 12.715	51.25 ± 14.083	<0.01	
KOOS quality of life	47.31 ± 11.954	74.81 ± 8.352	<0.01	
Knee flexion ROM (degrees)	119.13 ± 9.619	132.06 ± 7.663	<0.01	
Knee extension ROM (degrees)	−3.00 ± 3.759	−0.81 ± 1.559	<0.01	
Notes.

Abbreviations NPRS Numeric Pain Rating Scale (0-10)

KOOS Knee Injury and Osteoarthritis Outcome Score (0–100, higher = better)

ROM range of motion

ADL activities of daily living

QOL quality of life

Negative values in extension ROM indicate hyperextension.

Functional improvements showed similar patterns of benefit. The supervised group achieved markedly better scores across all KOOS domains: symptoms (86.38 ± 11.17 vs 62.50 ± 11.97), daily activities (81.94 ± 8.97 vs 52.75 ± 10.98), sports/recreation (51.25 ± 14.08 vs 23.75 ± 12.72), and quality of life (74.81 ± 8.35 vs 47.31 ± 11.95) at p-value < 0.01. These consistent advantages suggest that therapist-guided exercise not only enhances physical capacity but also helps patients regain confidence in their knee function during both basic and demanding activities.

Mobility outcomes further reinforced this pattern. While both groups showed improvement in range of motion, the supervised group achieved significantly greater gains in knee flexion (132.06° ± 7.66 vs 119.13° ± 9.62) and faster recovery of full extension (−0.81° ± 1.56 vs −3.00° ± 3.76) at p-value < 0.01. These differences likely reflect the value of professional mobilization techniques and precise form correction available in supervised settings.

According to the results of Table 6 supervised exercise program group consistently performed much better with comparison to the home-based group across all key outcomes features. Substantial improvements were seen in pain reduction, functional capacity (KOOS subscales), and knee mobility (ROM), representing that a structured as well as supervised rehabilitation strategy in post-PRP injections leading to target associated clinical outcomes in knee osteoarthritis patients.

Within-group analysis

Home-based exercise program group

There was a substantial improvement noticed in intensity of pain, KOOS pain, KOOS symptoms, knee flexion ROM, and knee extension ROM within the home-based exercise program group at (p < 0.05). Meanwhile, the changes in KOOS activities of day by day living, KOOS sports and recreation, as well as KOOS quality of life assessments were not statistically significant (p > 0.05), as indicted in Table 7.

Table 7 Within-group analysis for Group A (Home-Based) and Group B (Supervised) using paired samples t-test.

Between-group outcomes post-intervention.

Variable	Home-based exercise program group	Supervised exercise program group	
	Pre-intervention Mean ± SD	Post-intervention Mean ± SD	p-value	Pre-intervention Mean ± SD	Post-intervention Mean ± SD	p-value	
Pain intensity	6.81 ± 1.276	4.31 ± 0.873	0.001	6.94 ± 1.34	2.44 ± 0.964	0.001	
KOOS pain	47.50 ± 12.946	57.06 ± 8.330	0.001	51.50 ± 13.579	79.44 ± 10.551	0.001	
KOOS Symptoms	51.31 ± 14.664	62.50 ± 11.967	0.001	60.94 ± 22.062	86.38 ± 11.171	0.001	
KOOS activities of daily living	48.63 ± 12.816	52.75 ± 10.98	0.081	52.38 ± 12.961	81.94 ± 8.970	0.001	
KOOS sports and recreation	18.44 ± 11.933	23.75 ± 12.715	0.08	20.94 ± 12.546	51.25 ± 14.083	0.001	
KOOS Quality of Life	42.50 ± 16.725	47.31 ± 11.954	0.116	42.06 ± 16.275	74.81 ± 8.352	0.001	
Knee flexion ROM (degrees)	110.50 ± 12.681	119.13 ± 9.619	0.001	113.63 ± 11.407	132.06 ± 7.663	0.001	
Knee extension ROM (degrees)	−7.75 ± 5.447	−3.00 ± 3.759	0.001	−7.81 ± 5.256	−0.81 ± 1.559	0.001	
Notes.

Abbreviations NPRS Numeric Pain Rating Scale (0-10)

KOOS Knee Injury and Osteoarthritis Outcome Score (0–100, higher = better)

ROM range of motion

ADL activities of daily living

QOL quality of life

Negative values in extension ROM indicate hyperextension.

Supervised exercise program group

In the supervised exercise program group, statistically significant modifications were presented in approximately all the parameters, comprising the pain intensity, KOOS pain, KOOS symptoms, KOOS activities of day to day living, KOOS sports and recreation, KOOS quality of life, knee flexion ROM, and knee extension ROM (p < 0.05), as presented in Table 7.

In short, both interventional groups indicated remarkably innovations within their respective domains, but the supervised exercise program group more likely represented much improvements in reducing pain, functional output, knee ROM, and QOL with comparison to the home-based group. This signifies that a supervised rehabilitation program followed by PRP injections may be more target oriented in managing knee OA than an unsupervised home-based regimen.

Discussion

Our study demonstrates that supervised exercise following PRP injections yields superior clinical outcomes compared to home-based rehabilitation for knee osteoarthritis patients. Participants receiving guided therapy achieved greater improvements in pain relief, functional recovery, and knee mobility, with particularly notable benefits for activities of daily living and quality of life. While both approaches showed therapeutic value, the supervised program’s structured progression and professional oversight appear to amplify PRP’s regenerative potential, especially for restoring complex functional capacities beyond basic pain reduction.

The primary symptom of knee OA is pain and is most common leading factor that limits performance and quality of life of OA patients. In the current study, the supervised exercise group represented an increase in reduction in intensity of pain as per the findings from NPRS score with a comparison to the home-based group. This is consistent with prevailing literature suggesting the importance of structured, professionally guided exercise programs results in decrease in intensity of pain (Atukorala & Hunter, 2023; Bennell & Hinman, 2005; Filardo et al., 2015). The supervised exercises programs may provide best option for relief in pain because of more regulated, controlled, progressive regimen, appropriate guidance and form adjustments during workout, thereby reducing needless efforts and pain exacerbation.

Moreover, the KOOS pain subscale indicated that OA patients in the supervised group experienced a substantially higher performance in pain scores with comparison to the home-based group. This result further implement the hypothesis that supervised treatment protocols allow for the betterment of pain management, effectually because of inculcation of hands on techniques like knee mobilizations, which are not segmented as unsupervised home programs (Kohn, Sassoon & Fernando, 2016).

The secondary objective of the current study was to assesses and evaluate the functional performance enhancement and upgradation of standards of quality of life in OA patients. The KOOS subscales of day-to-day activities, sports as well as recreation, and QOL showed significant betterment in the exercise supervised group. Particularly, the ADL scores for day to day activities suggest that participants who involved in a supervised program restored an increase in performance of everyday tasks, like sitting, walking and stair climbing, which are most commonly suffered in knee OA patients (Laudy et al., 2015; Silverwood et al., 2015). Personalized feedback suggests a good score for supervised exercises as more beneficial because of lack in progression in pain intensity and difficulty of daily tasks, which helps in strengthening more effectively than unsupervised, hectic routines.

Sports and recreation subscale the KOOS reflects the tendency to involve in more versatile physical activities, also indicated significant betterment in the supervised group. As far as knee mobilizations are concerned the targeted strengthening exercises were important feature of the supervised regimen, it is not astonishing that participants were able to restore better control and mobility, allowing for enhanced participation in physical activities (Cross et al., 2014).

Moreover, the QOL score indicated that the improvement in supervised group was much better than the home-based group. In supervised program the patients were benefited from the comprehensive guide lines, encouragement, as well as its adherence to their rehabilitation plan, which took part to an overall improved sense of well-being. Most of the previous studies have same conclusion that supervised programs most commonly result in much better patient satisfaction and improvements in physical and psychological health of knee OA patients (McAlindon et al., 2014; Srikanth et al., 2005).

One of the prime clinical objectives of this current study was to examine changes in knee ROM, specifically flexion and extension. The results showed that both interventional groups revealed improvements in ROM but the supervised group represented more beneficent in both knee extension and flexion. Knee flexion ROM increases more in the supervised group, which may be credited to the manual mobilization techniques that were important component of their therapy sessions. These interventional techniques is used to restore joint mechanics, relieve muscles and joint stiffness, and enhance flexibility more successfully with comparison to self-directed home-based exercises (Fransen et al., 2008).

Knee extension ROM also showed considerable improvement in both groups but this improvement was more prominent in supervised group. This reveals that supervised guidance can optimize the recovery of both flexion as well as extension, which were the important components for functional activities such as walking, squatting, and standing (Hunter & Bierma-Zeinstra, 2019). ROM improve in the supervised group can be associated to a better adherence to the already structured exercise plan.

The results of this study revealed prominent evidence that supervised exercise programs offer better outcomes measures with comparison to the home-based group compared to home-based exercise programs for knee OA patients following PRP injections. While home-based programs are often offered for relaxation, convenience and cost effective. Home-based exercise programs are deficient of structured feedback, source of motivation, and advancement provided in supervised settings. Supervised programs implemented that exercises are done in its true sense with reduction of injury risks and promote better attachment to the rehabilitation regimen, leading to successive outcomes measures of pain relief, functional recovery, as well as ROM improvements.

Numerous studies back up the idea that supervised interventions, specifically those involving physical therapy and manual therapy techniques, have a more remarkable impact on functional recovery with comparison to the home-based and unsupervised programs (Deyle et al., 2005; Roddy et al., 2005). In this study the higher KOOS scores of supervised group reflect these findings and it also suggest that structured, professional guidance is important for highlighting recovery in knee OA patients.

Limitations

While this study provides valuable insights into post-PRP rehabilitation approaches, several limitations warrant consideration:

Our exclusive reliance on knee flexion/extension ROM measurements, while clinically relevant, does not fully capture functional performance deficits. The inclusion of standardized functional tests (e.g., 30-second sit-to-stand, timed up-and-go) and isokinetic strength measurements would have provided more comprehensive evaluation of lower extremity function. This limitation may affect the generalizability of our functional outcome interpretations.

The study did not assess critical biomechanical factors such as joint loading patterns, gait parameters, or muscle activation strategies that influence OA progression and treatment response. Future studies should incorporate motion analysis and electromyography to address this gap. Our 1-month follow-up precludes conclusions about the durability of observed benefits. A longer observation period with serial assessments would clarify whether between-group differences persist or converge over time. The nature of the interventions prevented blinding of participants and therapists to treatment allocation, potentially introducing performance bias. However, outcome assessors remained blinded to group assignment throughout the study. Our focus on moderate OA (KL grades 1–3) in middle-aged patients (40–60 years) may limit applicability to severe OA cases or older populations with greater comorbidities. Similarly, the absence of intermediate (3-month) or long-term (6-month) assessments limits our ability to determine whether: (1) initial group differences persist, or (2) home-based participants achieve comparable results with delayed progression. This represents a key direction for future research.

Clinical Implications

The results of this current study have important clinical practice implications. Supervised exercise programs, when combined with PRP injections, should be considered a primary strategy for managing knee OA, more specifically in patients who are searching for non-surgical treatment intervention. Healthcare personals should focus the value of designing structured rehabilitation and considering integration manual therapies and progressive strengthening exercises to improve patient outcomes. To strengthen the clinical relevance of these findings, future studies should incorporate follow-up assessments at 3 and 6 months. This extended timeline would clarify whether the observed short-term benefits of supervised exercise persist, diminish, or converge with home-based outcomes over time. Such data would better inform rehabilitation protocols by identifying optimal durations for supervised therapy and potential need for booster sessions. Additionally, serial evaluations could reveal whether PRP’s biological effects synergize differently with exercise modalities as the joint microenvironment evolves post-injection. Addressing this gap would provide critical guidance for long-term management of knee OA.

Conclusion

This study emphasizes that a supervised exercise program following PRP injections results in major improvements in pain intensity, ROM, as well as other functional outcomes with comparison to the home-based exercise program in patients with knee OA. The findings also highlight the importance of structured rehabilitation in optimizing recovery and improving the QOL for knee OA patients.

Supplemental Information

Supplemental Information 1 Raw data

Additional Information and Declarations

Competing Interests

Author Contributions

Human Ethics

Data Availability

The authors declare there are no competing interests.

Ghalia Safdar conceived and designed the experiments, prepared figures and/or tables, and approved the final draft.

Mubin Mustafa Kiyani conceived and designed the experiments, prepared figures and/or tables, and approved the final draft.

Salman Ahmed Saleem performed the experiments, prepared figures and/or tables, and approved the final draft.

Wajeeha Irfan Qureshi performed the experiments, prepared figures and/or tables, and approved the final draft.

Shahid Bashir analyzed the data, authored or reviewed drafts of the article, and approved the final draft.

Turki Abualait analyzed the data, authored or reviewed drafts of the article, and approved the final draft.

Hamid Khan performed the experiments, authored or reviewed drafts of the article, and approved the final draft.

Abdul Ghafoor Sajjad conceived and designed the experiments, authored or reviewed drafts of the article, and approved the final draft.

The following information was supplied relating to ethical approvals (i.e., approving body and any reference numbers):

Approval from IRB of Shifa International Hospital was taken before commencement of the study under the letter no 1080-356-2018.

The following information was supplied regarding data availability:

The raw data are available in the Supplemental File.

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
