# Peer review of "Comparison of supervised versus home-based exercise programs following platelet-rich plasma injections in knee osteoarthritis: clinical outcomes from quasi-experimental study"

_PeerJ, doi:10.7717/peerj.19971_

## Round 0.1 · original submission · Major Revisions

Reviewer 1 ·

Basic reporting

This quasi experimental study compared supervised training between home-based
exercises following PRP injections in Knee Osteoarthritis.
Here my comments:

MAIN TEXT: Please ensure consistent and rational use of capital letters (e.g., in the title, capitalization is acceptable but should follow a logical structure; in the abstract: use “characterized” with appropriate capitalization; in section 2.1: avoid unnecessary capitalization such as “Current”… and so long).

Experimental design

METHODS:
- Sections 2.6, 2.7, and 3.4 should be written in a more fluid and narrative style, avoiding very short subsections of only 2–6 lines.
- What tool or scale was used to assess the severity of osteoarthritis in your patients as inclusion criteria? You cited reference 7 regarding the Kellgren–Lawrence Classification, but it is not clear whether this system was actually applied in your study population. Please clarify.

TABLES:
Include in Table 1— or add an additional table or figure —information on the timing of treatments: when the injections began, how long the treatment lasted, and the schedule in relation to the rehabilitation sessions.
- Add a legend at the end of all tables to explain abbreviations and provide clarity.

Validity of the findings

RESULTS: compare the results at the beginning of the study to show the homogeneity of the samples

Additional comments

INTRODUCTION: It may be useful for improving the interest of the readers to introduce that other substances are commonly used in the treatment of knee osteoarthritis, often in combination with various rehabilitation approaches. I suggest adding for example that over the past two decades, local intra-articular knee injection therapy has gained popularity due to the advent of platelet-rich plasma (PRP), hyaluronic acid (HA), and the novel peripheral blood-derived mononuclear cells (PBMNCs) (DOI: 10.3390/jfmk10020104)."

DISCUSSION: Begin the discussion section by briefly summarizing the main findings of your results, without repeating specific statistical number values. Avoid restating the study aim at the beginning of this section.

Reviewer 2 ·

Basic reporting

The manuscript is generally satisfactory written. However, the design of this paper should be improved before publishing.
In my opinion, it is obligatory:
1. To present in Abstract (Line 30) information about the age range and sex (%females) of the knee OA patients recruited for better understanding the measured participants.

Experimental design

In my opinion, it is obligatory:
1. To use the term 'The main aim' instead of 'The main theme' at the end of 1. Introduction (Line 102) for indication of the objective of this study.
2. To describe more detailed the exclusion criteria of this study in 2. Materials and Methods, 2.5. Sample Selection. For example, how was controlled that the participants had no:
(1) Neurological impairment.
(2) History of systemic disease such as rheumatoid arthritis.
(3) Cardiovascular disease.
(4) Cognitive impairment.
The authors should answer also the question: „Did the participants received physiotherapy in last 6 months?“

4. To rewrite in 2. Materials and Methods paragraphs 2.6. Measurement Tool and 2.7. Outcome Measures, making one paragraph „Measures“ for better presenting the research methods used in this study and to avoid doublation in text. In this paragraph, the authors should clearly indicate:
(1) The type and producing company (country) of the goniometer used in this study.
(2) The type of knee extension and flexion ROM: active or passive.
(3) How was selected the trial for further analysis during testing the knee extension and flexion ROM (by best result or by mean of three sets).

5. To indicate in 2. Materials and Methods, 2.11. Data Analysis that:
(1) Data are presented as mean ± standard deviation (SD).
(2) The statistical significance was set at p<0.05.

Validity of the findings

I suggest:
1. To present more limiting factors of this study in 5. Limitations:
(1) Only knee extension and flexion ROM was used for objective measurement of knee function, which makes difficulties during interpretation of the results of the present study. Functional performance test such as knee extensor and flexor muscle strength, Timed Up and Go test, Sit to Stand test would give more information about the functional status of the knee OA patients.
(2) Only pre- and post-treatment measurement (a short-term effect of exercise therapy) was used in this study. I would be better to perform a measurement at 3 or 6 months to determine whether there is a long-term effect of exercise therapy with PRP injections in knee OA patients.

Additional comments

I suggest:
1. To delete the text in the beginning of the 3. Results (Lines 203-207) because this is only repetition of the main aim of this study and not connect the real results of the study.

2. To create a table in 3. Results, 3.1. Demographics and Baseline Characteristics. that illustrates the mean age, sex (%females) and anthropometric data (height, weight, body mass index) of the participants in two measured groups at baseline, with indication of statistical significance between the groups. In the present form, the information only for mean age of participants without statistics is too general.

3. To indicate in 3. Results, Table 4 that data are mean ± SD.

---

## Round 0.2 · accepted · Accept

Congratulations on the acceptance of your manuscript. The study addresses a highly relevant clinical topic, and your findings provide valuable insights into the role of supervised exercise in enhancing outcomes following PRP injections in knee osteoarthritis. The methodology is clearly presented, and the results are consistent with current evidence. A careful language revision is recommended to further improve clarity and scientific tone. Well done on your contribution to the field.

Reviewer 1 ·

Basic reporting

-

Experimental design

-

Validity of the findings

-

Additional comments

I appreciated the efforts of the authors to improve the manuscript. They followed the suggestions and ìncreased the interest of the readers.

Reviewer 2 ·

Basic reporting

-

Experimental design

-

Validity of the findings

-

Additional comments

The design of this manuscript was improved in the process of review.